# Association between Secondhand Smoke Exposure and Hypertension in 106,268 Korean Self-Reported Never-Smokers Verified by Cotinine

**DOI:** 10.3390/jcm8081238

**Published:** 2019-08-16

**Authors:** Byung Jin Kim, Jeong Gyu Kang, Ji Hye Kim, Dae Chul Seo, Ki Chul Sung, Bum Soo Kim, Jin Ho Kang

**Affiliations:** 1Division of Cardiology, Department of Internal Medicine, Kangbuk Samsung Hospital, Sungkyunkwan University School of Medicine, Seoul 03181, Korea; 2Center for Cohort Studies, Total Healthcare Center, Kangbuk Samsung Hospital, Sungkyunkwan University School of Medicine, Seoul 03181, Korea

**Keywords:** smoking, secondhand smoke, passive smoke, hypertension, population, epidemiology

## Abstract

No study has reported the relationship between secondhand smoke (SHS) exposure and hypertension in self-reported never-smokers verified by nicotine metabolite. The aim of this study is to determine the relationship between SHS exposure and hypertension in self-reported and cotinine-verified never-smokers. A total of 106,268 self-reported never-smokers, verified as nonsmokers by urinary cotinine, who participated in Kangbuk Samsung Cohort study (KSCS) between 2012 and 2016 were included. Cotinine-verified nonsmokers were defined as individuals having urinary cotinine <50 ng/mL. SHS exposure was defined as current exposure to passive smoke indoors at home or the workplace. The multivariate regression model revealed that SHS exposure was associated with hypertension (odds ratio (OR) (95% confidence interval (CI)), 1.16 (1.08, 1.24)). Current SHS exposure that has been exposed to home SHS (1.22 (1.11, 1.33)) as well as current SHS exposure only at the workplace (1.15 (1.02, 1.29)) significantly increased the ORs for hypertension compared to no SHS exposure. There was no significant gender interaction for the relationships between SHS exposure and hypertension. This study showed that SHS exposure was significantly associated with hypertension in self-reported never-smokers verified as nonsmokers by urinary cotinine, suggesting necessity of health program and stricter smoking regulation to reduce the risk of hypertension.

## 1. Introduction

Minimizing secondhand smoke (SHS) exposure has been a public health target globally. Although the prevalence of SHS exposure in Korea is decreasing with extension of the range of smoke-free public facilities since 2012, the prevalence is not yet low. The Korean National Health and Nutrition Examination Survey (KNHANES) VII-1 reported that the prevalence of SHS exposure indoors at home or at the workplace was 27.5% for male never-smokers and 20.0% for female never-smokers [1]. However, because information on cigarette smoking is based on self-reported smoking questionnaires, actual smoking status could not be exactly assessed; respondents overreport never smoking because smoking is considered a socially undesirable behavior. Cotinine has been used as a biomarker for the validation of self-reported smoking status. A previous study demonstrated that self-reported SHS significantly underestimated the actual SHS as determined by cotinine verification [2]. 

SHS exposure as well as active smoking are known risk factors for various cardiovascular diseases [3,4]. Meanwhile, the relationship between active smoking and hypertension somewhat showed discordant results in previous studies [5,6,7,8]. Recently, researchers on a number of studies have investigated the relationship between SHS exposure and hypertension, and most have found a positive relationship between the two [9,10,11,12,13,14,15]. However, the studies’ data have been based on self-reported questionnaires, and most studies used relatively small sample sizes or were restricted to only women. 

Therefore, the aim of this cross-sectional study was to evaluate the relationship between SHS exposure and hypertension among self-reported never-smokers verified by cotinine levels and to compare the relationship according to sex.

## 2. Methods

### 2.1. Study Population

The Kangbuk Samsung Cohort Study (KSCS) is an ongoing prospective cohort study that was conducted on adult Korean men and women who underwent a comprehensive annual or biennial examination at Total Healthcare Centers of the Kangbuk Samsung Hospital since 2012. For this study, we initially enrolled 108,354 self-reported never-smokers between 2012 and 2016. Among them, we excluded three individuals who did not measure urinary cotinine and 2083 individuals who had a urinary cotinine level of above 50 ng/mL to reduce misclassification error of self-reported smoking status. Finally, we included 106,268 self-reported never-smokers verified as nonsmokers by urinary cotinine in this study (Appendix A). This study was approved by the Institutional Review Board of Kangbuk Samsung Hospital (IRB No: 2018-06-029) and all participants provided written informed consent prior to their inclusion in the study.

### 2.2. Blood Pressure (BP) Measurements

The methodology of anthropometric and laboratory data has been previously described in detail [7,16]. Seated BP was measured three times at least two minutes apart by trained nurses with an automated oscillometric device (53000-E2, Welch Allyn, New York, NY, USA) after a 5 min rest. The 2nd and 3rd of the three measurements were averaged to obtain the systolic BP (SBP) and diastolic BP (DBP). We defined hypertension as elevated BP (SBP ≥ 140 mmHg and/or DBP ≥ 90 mmHg) or currently taking antihypertensive medication.

### 2.3. Assessment of Smoking Status

Based on standardized questionnaires, we defined individuals as self-reported never-smokers if they never smoked or had smoked fewer than five packs in their lives. Urine cotinine level was measured after 10 h of a smoking-free period with the DRI Cotinine Assay (Microgenics Corp., Fremont, CA, USA) using Modular P800 (Roche Diagnostics, Tokyo, Japan). The test had a detection limit of 34 ng/mL and all the tests were performed according to the manufacturer’s instructions. We defined cotinine-verified nonsmokers as individuals with urinary cotinine <50 ng/mL [17]. The questionnaire of secondhand smoke exposure is presented in Figure 1. We defined SHS exposure as current exposure to passive smoke indoors at home or the workplace. Home SHS exposure was defined as previous exposure to passive smoke at home regardless of current home and workplace SHS exposure, and divided into two groups: home SHS exposure before age 20 and home SHS exposure after age 20. Individuals with SHS exposure only in the workplace was defined as those who answered “yes” to the 3rd question of the questionnaire without home SHS exposure before and after age 20.

### 2.4. Statistical Analysis

Continuous variables were expressed as mean ± standard deviation or median (interquartile ranges). Categorical variables were expressed as percentages (%). Serum triglyceride (TG) and high-sensitivity C reactive protein (hsCRP) were log-transformed for analysis. Data in tables are expressed as untransformed original data for easy interpretation. We compared the characteristics of two groups according to presence/absence of SHS exposure and hypertension using Student’s *t-*test or Chi-square test. 

We conducted multivariate logistic regression analyses to assess the relationship between SHS exposure and hypertension. The multivariate model was adjusted for age, sex, waist circumference, body mass index, frequency of alcohol drinking, frequency of vigorous exercise, glucose, creatinine, uric acid, total cholesterol, high-density lipoprotein cholesterol, low-density lipoprotein cholesterol, TG, and hsCRP, which were variables with univariate relationships (*p* < 0.05). Furthermore, we used the multivariate model to determine the association between duration and frequency of SHS exposure and hypertension. We also performed multivariate linear regression analyses to assess the relationship between SHS exposure and BP after excluding individuals taking antihypertensive medication. We performed the statistical analyses using IBM SPSS version 24 (IBM Corp, Armonk, NY, USA) and we considered *p* < 0.05 statistically significant.

## 3. Results

### 3.1. Prevalence of SHS Exposure-Related Variables and Hypertension

The prevalence of SHS exposure in the overall population was 21.7%, with rates of 26.2% for males and 19.6% for females (*p* < 0.001). SHS exposure rate by year were 25.3% in 2012, 25.8% in 2013, 20.8% in 2014, 16.6% in 2015, and 15.1% in 2016. The overall prevalence of hypertension was 5.4%; the prevalence in males was higher than that in females (10.5% versus 3.2%, *p* < 0.001). 

### 3.2. Characteristics of the Two Groups According to Absence/Presence of SHS Exposure and Hypertension

The characteristics between the groups with and without SHS exposure and the two group according to the presence or absence of hypertension showed significant differences for all variables (Table 1 and Appendix A). In particular, the prevalence of hypertension in the group with SHS exposure was higher than that in the group without (6.7% versus 5.1%, *p* < 0.001). The group with hypertension showed higher prevalence of SHS exposure than did the group without (26.6% versus 21.4%, *p* < 0.001).

### 3.3. Association Between SHS Exposure and Hypertension in the Overall Population

The age-and sex-adjusted regression model revealed that SHS exposure was associated with hypertension (odds ratio (OR) (95% confidence interval (CI)), 1.23 (1.16, 1.31)) and the OR in the multivariate model remained significant (1.16 (1.08, 1.24); Table 2). The above results were similar to the results when we applied the definition of hypertension in the new ACC/AHA high blood pressure clinical practice guideline, ≥130/80 mmHg (Appendix A) [18]. The group with current SHS exposure that has been exposed to home SHS (1.22 (1.11, 1.33)) as well as the group with current SHS exposure only at the workplace (1.15 (1.02, 1.29)) significantly increased the ORs for hypertension compared to the group without current SHS exposure that has been not exposed to home SHS (Table 3). Furthermore, per 1 h increase in daily time, per 1 unit increase in frequency, and per 1 year in duration of SHS exposure significantly increased the risk of hypertension 3%, 5%, and 0.4%, respectively (Table 4). In particular, SHS exposure of <1 h/day, <3 times/week, and < 0 years increased the risk of hypertension 16–32% above that for individuals with no SHS exposure (1.32 (1.19, 1.46), 1.17 (1.08, 1.27), and 1.16 (1.02, 1.31), respectively). These results were similar to those in the group with SHS exposure of ≥1 h/day, ≥3 times/week, and ≥10 years (Table 4).

### 3.4. Association Between SHS Exposure and BP in Overall Population

In the age-and sex-adjusted linear regression model, SHS exposure in the overall population was significantly associated with SBP and DBP (regression coefficient (95% CI), 0.41(0.26, 0.56) for SBP and 0.39 (0.27, 0.51) for DBP, all *p* < 0.001; Table 5). However, the multivariate model showed that SHS exposure significantly increased only DBP (0.14 (0.02, 0.26); Table 5). The results of 104,201 individuals excluding 2067 individuals who were taking antihypertensive medication were similar to the above results (Table 5). 

### 3.5. Subgroup Analyses for the Relationship Between SHS Exposure and Hypertension According to Sex

Subgroup results according to sex for the relationships between SHS exposure and hypertension were similar, without significant gender interaction (*p* for interaction = 0.255). SHS exposure in both males and females was associated with hypertension in the multivariate regression model (1.14 (1.05, 1.24) for males and 1.15 (1.02, 1.30) for females; Table 2). 

## 4. Discussion

This study showed that SHS exposure in self-reported never-smokers verified by urinary cotinine was significantly associated with prevalence of hypertension. This association was proportional to the frequency and duration of SHS exposure. However, even the group with relatively low frequency and short duration of SHS exposure was significantly associated with hypertension compared to the group without SHS exposure. Furthermore, current SHS exposure only at the workplace as well as current SHS exposure that has been exposed to home SHS was associated with hypertension.

With the enactment of the Law for the National Health Promotion in 1995, Korea introduced some nonsmoking areas in public places, and since 2012, efforts have been made to minimize SHS exposure by expanding designated smoking cessation areas in public facilities and places. As a result, according to KNHANE VII-1 report, the rate of SHS exposure has been decreasing since 2012 [1], and our study also showed that the rate of SHS exposure decreased from 28.3% in 2011 to 15.1% in 2016.

Previous studies on acute effect of active smoke on BP have established that smoking temporarily increases BP [19,20]. The effect of chronic smoking on the rise in BP is also known to be mainly due to nicotine-mediated sympathetic activation [21]. However, several studies reported tolerance to the effect of nicotine on BP and the possibility of effect of smoking on decreasing BP via vasodilating mediators [22,23,24]. Due to these various biological properties of cigarette smoking, previous epidemiologic studies have shown inconsistent results on the relationship between chronic smoking and BP [5,6,7,8]. The mechanism linking SHS exposure to hypertension is not fully elucidated but is thought to be similar to the mechanism between active smoking and blood pressure. Previous investigators have reported that SHS exposure causes structural and functional alterations on arterial walls including endothelial dysfunction, increased arterial stiffness, and progression of atherosclerosis [19,20,22], and these changes could lead to higher BP. There have been epidemiologic studies on the relationship between chronic SHS exposure and hypertension in diverse population groups [9,10,11,12,13,14,15], and most of them demonstrated that SHS exposure was significantly associated with hypertension [9,10,11,12,13,14], which was consistent with our results. However, considering that individuals who were enrolled in our study had a low level of cotinine and SHS contains relatively higher concentrations of the toxic materials from side stream smoke, putative mechanism of linking of SHS exposure to BP could not exclude the effects of other oxidant gases and toxic chemicals other than nicotine in cigarette. Further pathophysiologic studies are needed to clarify the biologic actions.

However, there are several key differences between previous studies and our study. First, most of the studies were limited to females [9,10,11,12], and the sample sizes were small except for one study [12]. The results of the studies did show a positive association of SHS exposure with hypertension or BP, which is consistent with our results. A recent study in a large Chinese population also revealed a significant relationship between SHS exposure and hypertension, but the study’s investigation was limited to how husbands’ smoking affected their wives’ hypertension status [12]. Although the remaining three studies included both males and females [13,14,15], authors of only one study of 10,532 males and females investigated the association between SHS exposure and hypertension according to sex [13]. Those authors found that SHS exposure was associated with hypertension in females only. In contrast, the subgroup analyses of our study showed significant associations in males as well as females, which was consistent with the main results of our study. The main reason for the difference of result in males between the two studies could be the different sample sizes, although the authors of the previous study explained that the lack of association with hypertension in males could have been because the participants were younger or because of sex-dependent biologic effects and small sample size. Second, all of the above studies were conducted based on self-reported data on SHS exposure, whereas, for our study, we assessed SHS exposure with minimizing misclassification of never-smokers using a biomarker as well as a self-reported questionnaire. Studies with only self-reported questionnaires cannot evade recall bias. For instance, in a recent Korean study derived from KNHANES IV, respondents who had self-reported SHS exposure had significantly underestimated their actual SHS exposure as verified by cotinine [2]. Furthermore, the results of previous studies have reflected a misclassification between self-reported and biomarker-verified smoking status [25,26]. In our previous studies, we reported that the proportion of cotinine-verified current-smokers among self-reported never-smokers is 1.7–1.8% [7,16,27]. In short, the prevalence of self-reported never-smokers may be higher than that of actual never-smokers. Third, for the present study, we indirectly analyzed the relationship of home and workplace SHS exposure with hypertension. We found two significant results; SHS exposure only at workplace was associated with hypertension; irrespective of presence/absence of SHS exposure at workplace, SHS exposure at home was associated with hypertension. These findings suggest the importance of banning smoking at home as well as at the workplace. Fourth, the association between SHS exposure and hypertension increased with increasing daily time, week frequency and duration units of SHS exposure, although the ORs was not high. However, even the group with lower levels of frequency and duration of SHS exposure were also associated with hypertension, similar to the group with higher levels of frequency and duration of SHS exposure. This result suggests that it is necessary to avoid SHS smoke exposure to reduce the prevalence of hypertension. 

Cotinine, a main metabolite of nicotine, has been known to be a useful biomarker in assessing actual smoking status with long half-life of about 17 h. Until now, although there is no clear cutoff point for urinary cotinine in terms of verifying current smoking, 50 ng/mL is generally used as a reference cutoff [17]. In our previous study with 167,868 individuals, the urinary cotinine cutoff of 50 ng/mL provided sensitivity of 90%, specificity of 98.2%, positive predictability of 95.6%, and negative predictability of 95.9% for assessing smoking status according to self-reported questionnaire and cotinine verification [7]. In particular, high specificity is an important indicator for minimizing the misclassification of true never-smokers. Recently, one study of 2889 never-smokers reported that higher serum cotinine levels, an objective marker of SHS exposure, were associated with higher systolic BP and the presence of hypertension [28]. However, strictly speaking, self-reported never-smokers with high cotinine levels would be referred to as unobserved smokers rather than as exposed to SHS. As mentioned in our previous studies, unobserved smoking includes SHS and under-self-reporting of cigarette smoking [7,16]. In particular, some smokers in Korea tend to make false statements of being never-smokers on self-reported smoking-related questionnaires because of the social recognition that the formerly Confucian culture is reluctant to accept female smokers and that there are potential disadvantages of male workers’ job life and promotion. Notably, the current study is the first to include cotinine testing as well as self-reported smoking and SHS exposure questionnaire to minimize the misclassification of SHS exposure. 

As described in our previous studies [7,8], the present study has several limitations as well. First, because the design of this study is cross-sectional, we cannot confirm the causal relationship between SHS exposure confirmed by self-reported questionnaire and cotinine and hypertension. Second, most of the study population was middle-aged people who resided in metropolitan areas; therefore, they might not be representative of the overall Korean population. Third, because the questionnaire items in this study did not completely separate the SHS exposure at home and workplace, our results would not be able to pinpoint the effect of SHS exposure at home and workplace on hypertension, respectively. However, based on the questionnaire presented in Figure 1, we could divide the four groups of SHS exposure at home and workplace indirectly. Lastly, we did not have any information on the potential parameters that could affect nicotine metabolism such as genetic variations and drugs, and this study did not include dietary factors influencing hypertension. Nonetheless, our study is meaningful in that we are the first to conduct a large epidemiologic study to assess the association between SHS exposure and hypertension in never-smokers verified by self-reported questionnaire and cotinine.

## 5. Conclusions

In conclusion, this study showed that SHS exposure was significantly associated with hypertension in self-reported never-smokers with low urinary cotinine levels (<50 ng/mL). SHS exposure only at the workplace was associated with hypertension and SHS exposure at home irrespective of whether SHS exposure at the workplace was associated with hypertension. Even the group with lower levels of frequency and duration of SHS exposure were also associated with hypertension, similar to the group with higher levels of frequency and duration of SHS exposure. The present study’s findings suggest that more active antismoking programs in the home and public are needed to reduce the risk of hypertension. Furthermore, longitudinal studies will be required to clarify the relationship with development of hypertension.

## Figures and Tables

**Figure 1 jcm-08-01238-f001:**
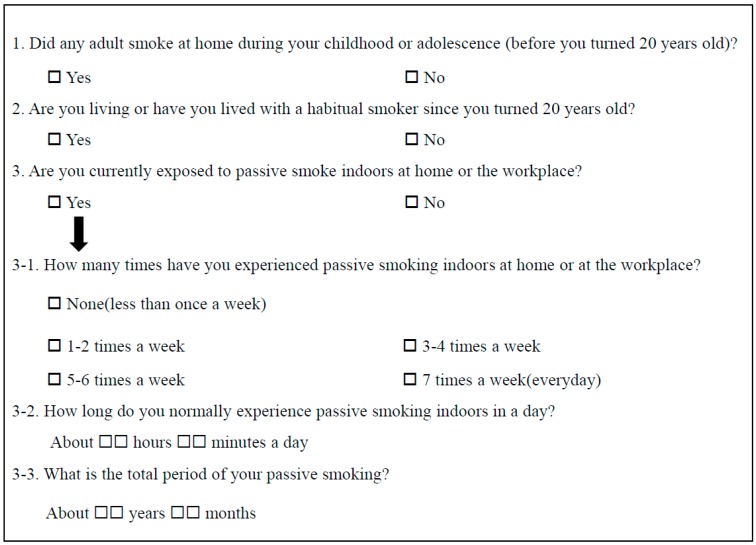
Questionnaire of secondhand smoke exposure.

**Table 1 jcm-08-01238-t001:** Characteristics of never-smokers with and without secondhand smoke exposure.

	SHS Exposure (−) *n* = 83,242	SHS Exposure (+) *n* = 23,026	*p* Value
Age, years	34.5 ± 7.1	34.1 ± 7.4	<0.001
Sex (males), *n* (%)	24,046/83,242 (28.9)	8550/23,026 (37.1)	<0.001
Body mass index, kg/m^2^	22.3 ± 3.3	22.7 ± 3.5	<0.001
Waist circumference, cm	77.8 ± 9.4	79.2 ± 9.9	<0.001
Systolic blood pressure, mmHg	104.1 ± 12.0	105.4 ± 12.3	<0.001
Diastolic blood pressure, mmHg	66.4 ± 9.0	67.4 ± 9.3	<0.001
Serum creatinine, umol/L	67.6 ± 16.2	70.6 ± 16.5	<0.001
Uric acid, umol/L	287.5 ± 80.4	299.3 ± 84.1	<0.001
Total cholesterol, mmol/L	4.84 ± 0.84	4.86 ± 0.84	<0.001
Triglyceride, mmol/L	0.84 (0.63, 1.16)	0.86 (0.63, 1.23)	<0.001
HDL cholesterol, mmol/L	1.63 ± 0.40	1.60 ± 0.40	<0.001
LDL cholesterol, mmol/L	2.93 ± 0.79	2.95 ± 0.80	0.004
Glucose, mmol/L	5.07 ± 0.60	5.11 ± 0.67	<0.001
hsCRP, mg/L	0.4 (0.2, 0.8)	0.4 (0.2, 0.8)	0.001
Vigorous exercise, *n* (%)			<0.001
none/week	55,566/82,334 (67.5)	13,716/22,782 (60.2)	
<3 times/week	17,638/82,334 (21.4)	6082/22,782 (26.7)	
≥3 times/week	9130/82,334 (11.1)	2984/22,782 (13.1)	
Daily alcohol consumption, gram/day	3.0 (1.0, 7.0)	6.0 (2.0, 14.0)	<0.001
Alcohol consumption, *n* (%)			<0.001
none/week	13,220/79,834 (16.6)	2492/22,261 (11.2)	
1–2 times/week	61,299/79,834 (76.8)	17,157/22,261 (77.1)	
3–4 times/week	4560/79,834 (5.7)	2265/22,261 (10.2)	
≥ 5 times/week	755/79,834 (0.9)	347/22,261 (1.6)	
Daily times of SHS exposure, *n* (%)			<0.001
None	83,242/83,242 (100)	0/13,528 (0)	
<1 h/day	0/83,242 (0)	9216/13,528 (68.1)	
≥1 h/day	0/83,242 (0)	4312/13,528 (31.9)	
Frequency of SHS exposure, *n* (%)			<0.001
None	83,171/83,171 (100)	0/22,757 (0)	
<3 times/wee	0/83,171 (0)	14,925/22,757 (65.6)	
≥3 times/week	0/83,171 (0)	7832/22,757 (34.4)	
Duration of SHS exposure, *n* (%)			<0.001
None	82,923/82,923 (100)	0/15,777 (0)	
<10 years	0/82,923 (0)	6418/15,777 (40.7)	
≥10 years	0/82,923 (0)	9359/15,777 (59.3)	
Home SHS before age 20, *n* (%)	38,453/83,242 (46.2)	13,174/23,026 (57.2)	<0.001
Home SHS after age 20, *n* (%)	19,466/82,756 (23.5)	10,288/22,844 (45.0)	<0.001
Hypertension, *n* (%)	4222/83,103 (5.1)	1533/22,974 (6.7)	<0.001

Data are shown as mean ± standard deviation, median (interquartile) or percentage. Triglyceride, hsCRP, and daily alcohol amount were log-transformed for this analysis. *p* values were based on Student’s *t-*test or Chi-square test. Secondhand smoke exposure was defined as current exposure to passive smoking indoors at home or the workplace. SHS, secondhand smoke; *n*, number; HDL, high-density lipoprotein; LDL, low-density lipoprotein; hsCRP, high-sensitivity C-reactive protein.

**Table 2 jcm-08-01238-t002:** Multivariate logistic regression analyses for the association of hypertension with secondhand smoke exposure in never-smokers.

	Cases of Hypertension ^a^	Odds Ratios (95% Confidence Interval)	
		Age-Adjusted	Multivariate
Overall ^b^			
SHS exposure (−)	4222/83,103 (5.1)	1	1
SHS exposure (+) ^c^	1533/22,974 (6.7)	1.23 (1.16, 1.31)	1.16 (1.08, 1.24)
Males			
SHS exposure (−)	2434/24,019 (10.1)	1	1
SHS exposure (+) ^c^	990/8532 (11.6)	1.20 (1.11, 1.30)	1.14 (1.05, 1.24)
Females			
SHS exposure (−)	1788/59,084 (3.0)	1	1
SHS exposure (+) ^c^	543/14,442 (3.8)	1.25 (1.12, 1.39)	1.15 (1.02, 1.30)

^a^ Values are expressed as n/N (%), where n is number of cases with a given variable and N is total number considered for that variable. ^b^ Models adjusted for sex in overall population. ^c^ Reference group is group without SHS exposure. Multivariate model was adjusted for age, waist circumference, body mass index, frequency of alcohol drinking, frequency of vigorous exercise, glucose, creatinine, uric acid, total cholesterol, high-density lipoprotein cholesterol, low-density lipoprotein cholesterol, triglyceride, and high-sensitivity C-reactive protein. SHS, secondhand smoke.

**Table 3 jcm-08-01238-t003:** Multivariate logistic regression analyses for the association of hypertension with type of secondhand smoke exposure in never-smokers.

	Cases of Hypertension ^a^	Odds Ratios (95% Confidence Interval)	
		Age-and Sex-Adjusted	Multivariate
No current SHS exposure that has been not exposed to home SHS	1861/38,147 (4.9)	1	1
No current SHS exposure that has been exposed to home SHS ^b^	2340/44,477 (5.3)	1.13 (1.05, 1.21)	1.07 (0.99, 1.14)
Current SHS exposure only at the workplace ^b^	465/6678 (7.0)	1.19 (1.07, 1.33)	1.15 (1.02, 1.29)
Current SHS exposure that has been exposed to home SHS ^b^	1057/16,115 (6.6)	1.37 (1.27, 1.49)	1.22 (1.11, 1.33)

^a^ Values are expressed as n/N (%), where n is number of cases with a given variable and N is total number considered for that variable. ^b^ Reference group is group without current SHS exposure that has been not exposed to home SHS. Multivariate model was adjusted for age, sex, waist circumference, body mass index, frequency of alcohol drinking, frequency of vigorous exercise, glucose, creatinine, uric acid, total cholesterol, high-density lipoprotein cholesterol, low-density lipoprotein cholesterol, triglyceride, and high-sensitivity C-reactive protein. SHS, secondhand smoke.

**Table 4 jcm-08-01238-t004:** Multivariate logistic regression analyses for the association of hypertension with frequency and duration of secondhand smoke exposure in never-smokers.

	Cases of Hypertension ^a^	Odds Ratios (95% Confidence Interval)
Daily time of SHS exposure		
None	4222/83,103 (5.1)	1
<1 h ^b^	582/9211 (6.3)	1.32 (1.19, 1.46)
≥1 h ^b^	324/4302 (7.5)	1.19 (1.04, 1.36)
Daily time of SHS exposure (per 1 h increase)		1.03 (1.01, 1.06)
Frequency of SHS exposure		
None	4213/83,032 (5.1)	1
<3 times/week ^b^	944/14,891 (6.3)	1.17 (1.08, 1.27)
≥3 times/week ^b^	563/7814 (7.2)	1.13 (1.02, 1.25)
Frequency of SHS exposure ^c^ (per 1 unit increase)		1.05 (1.02, 1.07)
Duration of SHS exposure		
None	4190/82,786 (5.1)	1
<10 years ^b^	365/6409 (5.7)	1.16 (1.02, 1.31)
≥10 years ^b^	688/9339 (7.4)	1.18 (1.07, 1.30)
Duration of SHS exposure (per 1 year increase)		1.00 (1.00, 1.01) ^d^

^a^ Values are expressed as n/N (%), where n is number of cases with a given variable and N is total number considered for that variable. ^b^ Reference group is group without SHS exposure. ^c^ Frequency of SHS exposure is the ordinal variable according to 3-1 questionnaire in Figure 1. ^d^
*p* = 0.024. The model was adjusted for age, sex, waist circumference, body mass index, frequency of alcohol drinking, frequency of vigorous exercise, glucose, creatinine, uric acid, total cholesterol, high-density lipoprotein cholesterol, low-density lipoprotein cholesterol, triglyceride, and high-sensitivity C-reactive protein. SHS, secondhand smoke.

**Table 5 jcm-08-01238-t005:** Multivariate linear regression analyses for the association of blood pressure with secondhand smoke exposure in never-smokers including and excluding individuals with antihypertensive medication.

	Regression Coefficient (95% Confidence Interval)	
	Age-Adjusted	Multivariate
**Overall population (*n* = 106,268)**		
**Systolic blood pressure**		
SHS exposure (+)		
Overall ^a^	0.41 (0.26, 0.56)	0.04 (−0.10, 0.19)
Males	0.45 (0.18, 0.71)	0.08 (−0.17, 0.33)
Females	0.39 (0.21, 0.58)	0.03 (−0.14, 0.22)
**Diastolic blood pressure**		
SHS exposure (+)		
Overall ^a^	0.39 (0.27, 0.51)	0.14 (0.02, 0.26)
Males	0.38 (0.17, 0.59)	0.13 (−0.07, 0.34)
Females	0.39 (0.25, 0.54)	0.15 (0.01, 0.30)
**Population excluding those with antihypertensive medication (*n* = 104,201)**		
**Systolic blood pressure**		
SHS exposure (+)		
Overall ^a^	0.37 (0.22, 0.52)	0.02 (−0.12, 0.17)
Males	0.48 (0.21, 0.74)	0.10 (−0.15, 0.35)
Females	0.33 (0.15, 0.51)	−0.01 (−0.18, 0.17)
**Diastolic blood pressure**		
SHS exposure (+)		
Overall ^a^	0.37 (0.25, 0.49)	0.13 (0.02, 0.25)
Males	0.38 (0.16, 0.59)	0.13 (−0.08, 0.33)
Females	0.37 (0.22, 0.51)	0.14 (−0.01, 0.28)

^a^ Models adjusted for sex in overall population. The multivariate model was adjusted for age, waist circumference, body mass index, frequency of alcohol drinking, frequency of vigorous exercise, glucose, creatinine, uric acid, total cholesterol, high-density lipoprotein cholesterol, low-density lipoprotein cholesterol, triglyceride, and high-sensitivity C-reactive protein. SHS, secondhand smoke.

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
