# Peer review of "Association between Secondhand Smoke Exposure and Hypertension in 106,268 Korean Self-Reported Never-Smokers Verified by Cotinine"

_jcm, 2019, doi:10.3390/jcm8081238_

Round 1

Reviewer 1 Report

This is a large study on never-smokers by history verified as non-smokers by urinary cotinine. In the abstract this should be made clear, because never-smokers cannot be defined by cotinine levels. Cotinine defines current non-smokers and also the authors said that participants were asked to stay smoke-free for 10 hours before the urine sampling.

According to the tables (one supplementary figure and 2 supp. tables were missing) only current SHS was related to the frequency of hypertension (defined as >140/90 or antihypertensive medication; supp. table 2 for >130/80 is missing). SHS before or after age 20 or its combination with current SHS exposure was not related to blood pressure, except for a sentence in line 221-222, which is not supported by data.

In addition to the significant increase of hypertension rates in the group with SHS exposure, a dose-response was found for SHS with hours/day, times/week and years of duration (line 139-141, table 4). Remarkable are significant increases of hypertension rates with SHS <1hr/day, or <3 times/week, or after <10 years duration (presumably not in combination?).

Age-adjusted blood pressure rates in non-smokers with and without exclusion of persons with antihypertensive medication were associated with current SHS exposure, but after full adjustment only diastolic BP in females stayed significant and after exclusion of antihypertensive medication in both sexes combined.

In the discussion the authors mention limitations from cross-sectional approach, possible selection bias (middle-aged urban population willing to participate) and possible confounders (nutrition), but it stays unclear if no attempt has been made to relate cotinine level to blood pressure or prevalence of hypertension or if this was not successful. 

Author Response

Response to Reviewer 1

We appreciate your valuable comments and are very happy to reply on them.

For your convenience, the same order of the point-by-point list as in your criticism-comments of first submission-was used in our response. Also, modification to the revised manuscript are expressed in red color.

This is a large study on never-smokers by history verified as non-smokers by urinary cotinine. In the abstract this should be made clear, because never-smokers cannot be defined by cotinine levels. Cotinine defines current non-smokers and also the authors said that participants were asked to stay smoke-free for 10 hours before the urine sampling.

Answer. Thank you for your valuable comment. As you mentioned, we have revised the definition of the participants enrolled in our study more clearly to avoid confusion in the Abstract and Method sections, as the follows.

‘A total of 106,268 self-reported and cotinine-verified never-smokers’ was corrected to ‘A total of 106,268 self-reported never-smokers verified as non-smokers by urinary cotinine’.

Also, ‘cotinine-verified never-smokers’ in line 30 was changed to ‘cotinine-verified non-smokers’.

‘self-reported and cotinine-verified never-smokers’ in line 38-39 was corrected to ‘self-reported never-smokers verified as non-smokers by urinary cotinine’.

‘self-reported and cotinine-verified never-smokers’ in line 74-75 was revised to ‘self-reported never-smokers verified as non-smokers by urinary cotinine’.

‘cotinine-verified never-smokers’ in line 91 was changed to ‘cotinine-verified non-smokers’.

According to the tables (one supplementary figure and 2 supp. tables were missing) only current SHS was related to the frequency of hypertension (defined as >140/90 or antihypertensive medication; supp. table 2 for >130/80 is missing). SHS before or after age 20 or its combination with current SHS exposure was not related to blood pressure, except for a sentence in line 221-222, which is not supported by data.

Answer. Thank you for your thoughtful comments. We submitted a supplementary figure and two supplementary tables together when submitting our 1st manuscript to this journal. We can see them on the e-submission page. We will show the figure and two tables here.

Also, as you mentioned the above, we totally agree with your comment of ‘SHS before or after age 20 or its combination with current SHS exposure was not related to blood pressure, except for a sentence in line 221-222, which is not supported by data.’ We deleted the sentence of ‘In addition, no current SHS exposure with home SHS exposure before age 20 showed a trend toward hypertension.’ in the Discussion section (line 220-222).

Figure S1.

Supplementary Figure legends

Figure S1. Flow chart for sample selection criteria

SHS, secondhand smoke

Table S1. Characteristics of never-smokers with and without hypertension

Hypertension (-)

n=100,322

Hypertension (+)

n=5,755

p value

Age, years

34.0±6.6

41.1±11.8

<0.001

Sex (males), n(%)

29,127/100,322(29.0)

3,424/5,755(59.5)

<0.001

Body mass index, kg/m2

22.2±3.2

25.6±4.0

<0.001

Waist circumference, cm

77.6±9.2

87.1±10.4

<0.001

Systolic blood pressure, mmHg

103.1±10.7

125.5±15.0

<0.001

Diastolic blood pressure, mmHg

65.7±8.0

81.9±12.0

<0.001

Serum creatinine, umol/L

67.7±15.3

77.9±27.1

<0.001

Uric acid, umol/L

287.1±79.7

343.0±91.3

<0.001

Total cholesterol, mmol/L

4.82±0.83

5.14±0.94

<0.001

Triglyceride, mmol/L

0.82[0.62, 1.15]

1.20[0.85, 1.75]

<0.001

HDL cholesterol, mmol/L

1.63±0.40

1.43±0.38

<0.001

LDL cholesterol, mmol/L

2.92±0.78

3.30±0.87

<0.001

Glucose, mmol/L

5.05±0.56

5.56±1.16

<0.001

hsCRP, mg/L

0.4[0.2, 0.7]

0.6[0.3, 1.3]

<0.001

Vigorous exercise, n(%)

<0.001

none/week

65,852/99,279(66.3)

3,300/5,647(58.4)

< 3 times/week

22,168/99,279(22.3)

1,508/5,647(26.7)

³ 3 times/week

11,269/99,279(11.3)

839/5,647(14.9)

Daily alcohol consumption, gram

4.0[1.0, 10.0]

5.0[1.0, 14.3]

<0.001

Alcohol consumption, n(%)

<0.001

none/week

14,795/96481(15.3)

887/5,442(16.3)

1-2 times/week

74,428/96,481(77.1)

3898/5,442(71.6)

3-4 times/week

6,248/96,481(6.5)

567/5,442(10.4)

³5 times/week

1,010/96,481(1.0)

90/5,442(1.7)

Daily times of SHS exposure, n (%)

<0.001

    None

78,881/91,488(86.2)

4,222/5,128(82.3)

    < 1 hour/day

8,629/91,488(9.4)

582/5,128(11.3)

   ³ 1 hour/day

3,978/91,488(4.3)

324/5,128(6.3)

Frequency of SHS exposure, n (%)

<0.001

    None

78,819/100,017(78.8)

4,213/5,720(73.7)

< 3 times/week

13,947/100,017(13.9)

944/5,720(16.5)

   ³ 3 times/week

7,251/100,017(7.2)

563/5,720(9.8)

Duration of SHS exposure, n (%)

<0.001

    None

78,596/93,291(84.2)

4,190/5,243(79.9)

    < 10 years

6,044/93,291(6.5)

365/5,243(7.0)

   ³ 10 years

8,651/93,291(9.3)

688/5,243(13.1)

Home SHS before age 20, n (%)

48,626/100,322(48.5)

2,910/5,755(50.6)

0.002

Home SHS after age 20, n (%)

27,987/99,694(28.1)

1,706/5,723(29.8)

0.005

SHS exposure, n (%)

21,441/100,322(21.4)

1,533/5,755(26.6)

<0.001

Ani-hypertensive medication(s), n (%)

0/100,322(0)

2,060/5,755(35.8)

<0.001

Data are shown as mean ± standard deviation, median[interquartile] or percentage.

Triglyceride, hsCRP, and daily alcohol amount were log-transformed for this analysis.

P values were based on Student’s t-test or Chi-square test.

Secondhand smoke exposure was defined as current exposure to passive smoking indoors at home or the workplace.

SHS, secondhand smoke; N, number; HDL, high-density lipoprotein; LDL, low-density lipoprotein; hsCRP, high-sensitivity C-reactive protein.

Table S2. Multivariate logistic regression analyses for the association of hypertension a with secondhand smoke exposure in never-smokers

Odds ratios[95% Confidence interval]

Cases of hypertension b

Age-adjusted

Multivariate

Overall c

SHS exposure (-)

9,083/83,103 (10.9)

1

1

SHS exposure (+) d

3,153/22,974 (13.7)

1.17[1.11, 1.22]

1.09[1.03, 1.14]

Males

SHS exposure (-)

5,518/24,019 (23.0)

1

1

SHS exposure (+) d

2,118/8,532 (24.8)

1.13[1.06, 1.20]

1.06[1.00, 1.13]

Females

SHS exposure (-)

3,565/59,084 (6.0)

1

1

SHS exposure (+) d

1,035/14,442 (7.2)

1.21[1.12, 1.31]

1.12[1.02, 1.21]

a Hypertension was defined as systolic/diastolic blood pressure ³ 130/80 mmHg using 2017 ACC/AHA high blood pressure clinical practice guideline.18

b Values are expressed as n/N (%), where n is number of cases with a given variable and N is total number considered for that variable.

c Models adjusted for sex in overall population

d Reference group is group without SHS exposure.

Multivariate model was adjusted for age, waist circumference, body mass index, frequency of alcohol drinking, frequency of vigorous exercise, glucose, creatinine, uric acid, total cholesterol, high-density lipoprotein cholesterol, low-density lipoprotein cholesterol, triglyceride, and high-sensitivity C-reactive protein.

SHS, secondhand smoke

In addition to the significant increase of hypertension rates in the group with SHS exposure, a dose-response was found for SHS with hours/day, times/week and years of duration (line 139-141, table 4). Remarkable are significant increases of hypertension rates with SHS <1hr/day, or <3 times/week, or after <10 years duration (presumably not in combination?).

Answer. Thank you for your remark. As shown in Table 4 and the Discussion section(line 223-228), our study showed that the association between SHS exposure and hypertension increased with increasing daily time, week frequency and duration units of SHS exposure, although the ORs was not high. However, even the group with lower levels of frequency and duration of SHS exposure were also associated with hypertension, similar to the group with higher levels of frequency and duration of SHS exposure. We think it looks threshold pattern. So, we mentioned ‘this result suggests that it is necessary to avoid SHS smoke exposure to reduce the prevalence of hypertension’ in Discussion section(line 227-228).

Age-adjusted blood pressure rates in non-smokers with and without exclusion of persons with antihypertensive medication were associated with current SHS exposure, but after full adjustment only diastolic BP in females stayed significant and after exclusion of antihypertensive medication in both sexes combined.

Answer. We appreciate your comment. We agree with your opinion. Our results shows that only DBP remained significant in the multivariate model and also this result was evident in female. However, there was no gender interaction in our study. Accordingly, the results may be difficult to interpret as SHS exposure affecting DBP only in women. Also, previous studies on the effects of SHS exposure on BP(SBP and DBP) also show conflicting results. Furthermore, because the design of our study was cross-sectional, further longitudinal study would be necessary to more clarify this association. We mentioned the contents in the Limitation and Conclusion sections.

In the discussion the authors mention limitations from cross-sectional approach, possible selection bias (middle-aged urban population willing to participate) and possible confounders (nutrition), but it stays unclear if no attempt has been made to relate cotinine level to blood pressure or prevalence of hypertension or if this was not successful. 

Answer. Thank you for your comments. We agree with you. In particular, because our study was cross-sectional, it would unclear whether SHS exposure is a risk factor for developing hypertension or elevating blood pressure. Therefore, we stated the necessity of longitudinal study in the Conclusion section.

I sincerely appreciate you again for your valuable comments.

Reviewer 2 Report

This is interesting study with a massive sample size, I  have no suggestions other than one more round of typos checking. 

Author Response

Reviewer 2 comments

This is interesting study with a massive sample size, I have no suggestions other than one more round of typos checking. 

Answer. We deeply appreciate your time for reviewing our study.

Reviewer 3 Report

This is an interesting study that seeks to establish the relationship between exposure to second hand smoke and elevated BP in a large group of individuals. It builds on previous work to better examine any effects of gender on response and to control for current smokers misreporting as non-smokers. There are several aspects of the study methods which would benefit from further explanation.

Line 80. Please define whether there was a minimum time between BP measurements?

Line 91. Second hand smoke exposure was self-reported. Was there a minimum exposure to SHS required to be considered as fitting this definition? If so please specify. How is SHS expected to affect Cotinine levels?

Line 92-94. The definition of SHS status is confusing. The text indicates it relates to current exposure but then goes on to define home exposure as exposure to passive smoke at home before and after age 20 regardless of current and exposure in the workplace. Please clarify? 

Line 99. Please define serum TG and hsCRP on first use

Author Response

Response to Reviewer 3 comments

We appreciate your valuable comments and are very happy to reply on them.

For your convenience, the same order of the point-by-point list as in your criticism-comments of first submission-was used in our response. Also, modification to the revised manuscript are expressed in red color.

This is an interesting study that seeks to establish the relationship between exposure to second hand smoke and elevated BP in a large group of individuals. It builds on previous work to better examine any effects of gender on response and to control for current smokers misreporting as non-smokers. There are several aspects of the study methods which would benefit from further explanation.

Line 80. Please define whether there was a minimum time between BP measurements?

Answer. Thank you for your comment. In our study’s protocol, a minimum time between BP measurements was 2 minutes. We have added it in Line 79.

“ Seated BP was measured three times at least minutes apart by trained nurses with…”

Line 91. Second hand smoke exposure was self-reported. Was there a minimum exposure to SHS required to be considered as fitting this definition? If so please specify. How is SHS expected to affect Cotinine levels?

Answer. Thank you for your valuable comment. Theoretically, persons with very heavy secondhand smoke exposure may increase cotinine levels. However, there is currently no accurate data on cotinine levels in persons with SHS exposure. Cotinine level would be different depending on whether SHS exposure is indoors or outdoors, and when cotinine tests are measured. In our cohort, among self-reported never-smokers, individuals with high cotinine level(especially very high) in those without self-reported SHS exposure existed, even though it’s a very small percentage (1.2%). Therefore, our study enrolled only individuals with a cotinine level of less than 50 ng/mL, which generally suggested to minimize misclassification of smoke exposure including SHS exposure. Accordingly, the definition of SHS exposure was based on self-reported questionnaires.

Line 92-94. The definition of SHS status is confusing. The text indicates it relates to current exposure but then goes on to define home exposure as exposure to passive smoke at home before and after age 20 regardless of current and exposure in the workplace. Please clarify? 

Answer. Thank you for your helpful comment. To be exact, Home SHS exposure is divided into two groups; Home SHS exposure before age 20 was defined as previous exposure to passive smoke at home before age 20 regardless of current home and workplace SHS exposure; home SHS exposure after age 20 was defined as previous exposure to passive smoke at home after age 20 regardless of current home and workplace SHS exposure. We have changed the sentence to the following:

“Home SHS exposure was defined as previous exposure to passive smoke at home regardless of current home and workplace SHS exposure, and divided into two groups: home SHS exposure before age 20 and home SHS exposure after age 20.”

Line 99. Please define serum TG and hsCRP on first use.

Answer. Thank you for your remark. We have changed them.

“triglyceride (TG) and high-sensitivity C-reactive protein (hsCRP)”

We really appreciate your valuable comments again.